# β-Cyclodextrin Nanophotosensitizers for Redox-Sensitive Delivery of Chlorin e6

**DOI:** 10.3390/molecules28217398

**Published:** 2023-11-02

**Authors:** Jaewon Jo, Ji Yoon Kim, Je-Jung Yun, Young Ju Lee, Young-IL Jeong

**Affiliations:** 1Gwangju Center, Korea Basic Science Institute, Gwangju 61186, Republic of Korea; jawon1119@kbsi.re.kr (J.J.); jykim1018@kbsi.re.kr (J.Y.K.); 2School of Chemical Engineering, Chonnam National University, Gwangju 61186, Republic of Korea; 3Research Center for Environmentally Friendly Agricultural Life Sciences, Jeonnam Bioindustry Foundation, Jeonnam 58275, Republic of Korea; jjyoung4@hanmail.net; 4Department of Dental Materials, College of Dentistry, Chosun University, Gwangju 61452, Republic of Korea; 5Tyros Biotechnology Inc., 75 Kneeland St. 14 Floors, Boston, MA 02111, USA

**Keywords:** β-cyclodextrin, redox-sensitive, cancer targeting, photodynamic therapy, chlorin e6

## Abstract

The aim of this study is to prepare redox-sensitive nanophotosensitizers for the targeted delivery of chlorin e6 (Ce6) against cervical cancer. For this purpose, Ce6 was conjugated with β-cyclodextrin (bCD) via a disulfide bond, creating nanophotosensitizers that were fabricated for the redox-sensitive delivery of Ce6 against cancer cells. bCD was treated with succinic anhydride to synthesize succinylated bCD (bCDsu). After that, cystamine was attached to the carboxylic end of bCDsu (bCDsu-ss), and the amine end group of bCDsu-ss was conjugated with Ce6 (bCDsu-ss-Ce6). The chemical composition of bCDsu-ss-Ce6 was confirmed with ^1^H and ^13^C NMR spectra. bCDsu-ss-Ce6 nanophotosensitizers were fabricated by a dialysis procedure. They formed small particles with an average particle size of 152.0 ± 23.2 nm. The Ce6 release rate from the bCDsu-ss-Ce6 nanophotosensitizers was accelerated by the addition of glutathione (GSH), indicating that the bCDsu-ss-Ce6 nanophotosensitizers have a redox-sensitive photosensitizer delivery capacity. The bCDsu-ss-Ce6 nanophotosensitizers have a low intrinsic cytotoxicity against CCD986Sk human skin fibroblast cells as well as Ce6 alone. However, the bCDsu-ss-Ce6 nanophotosensitizers showed an improved Ce6 uptake ratio, higher reactive oxygen species (ROS) production, and phototoxicity compared to those of Ce6 alone. GSH addition resulted in a higher Ce6 uptake ratio, ROS generation, and phototoxicity than Ce6 alone, indicating that the bCDsu-ss-Ce6 nanophotosensitizers have a redox-sensitive biological activity in vitro against HeLa human cervical cancer cells. In a tumor xenograft model using HeLa cells, the bCDsu-ss-Ce6 nanophotosensitizers efficiently accumulated in the tumor rather than in normal organs. In other words, the fluorescence intensity in tumor tissues was significantly higher than that of other organs, while Ce6 alone did not specifically target tumor tissue. These results indicated a higher anticancer activity of bCDsu-ss-Ce6 nanophotosensitizers, as demonstrated by their efficient inhibition of the growth of tumors in an in vivo animal tumor xenograft study.

## 1. Introduction

Photodynamic therapy (PDT) is believed to be a safe option for the treatment of cancer [1,2,3]. PDT, which is composed of light, oxygen, and photosensitizers, can be specifically applied to tumor tissue only, minimizing adverse physiological effects against normal cells or tissues. This is because photosensitizers are only activated and produce an excess amount of reactive oxygen species (ROS) in the field of light irradiation, eradicating abnormal cells [4,5]. However, PDT is suitable for epithelial or squamous cancer phenotypes such as cervical cancer, melanoma, and oral cancers since the light penetration depth is normally limited to 15 mm of the physiological interface [6,7,8]. From these intrinsic properties, PDT is a suitable treatment option for cervical cancers because the morphology of cervical cancer is mostly of a squamous cell carcinoma phenotype, and the irradiated light can easily penetrate the tumor tissues [9]. Furthermore, photosensitizers have little toxicity against normal cells or tissues in the absence of light irradiation, and, thus, side effects can be minimized. In contrast, chemotherapeutic agents have serious side effects such as bone marrow depression, neurotoxicity, hematological toxicity, neutropenia, and nephrotoxicity [9,10]. Despite these advantages of a PDT regimen, the use of PDT with traditional photosensitizers has limitations in clinical application because some of them, such as 5-aminolevulinic acid (5-ALA), have low tumor specificity, rapid clearance from the human body, and low penetration depth against tissues, and they then become distributed throughout the whole body [11,12]. Furthermore, these problems induce light-sensitive problems in patients [13]. Some photosensitizers, such as chlorin e6 (Ce6), have a low aqueous solubility, and the drug resistance against cancer cells is also a drawback for their clinical application [14,15].

To solve these problems, various vehicles such as nanoparticles, polymer conjugates, proteins, polymeric micelles, nanomaterials, and cyclodextrins have been studied [16,17,18,19,20]. Nano-dimensional carriers have various advantages such as long blood circulation, solubilization of hydrophobic drugs, active/passive targeting of tumors, avoidance of reticuloendothelial system uptake, and site-specific delivery of bioactive agents [21,22,23]. For example, Lin et al. reported that Ce6-encapsulated poly(lactide-co-glycolide) nanoparticles show sustained drug release properties over 3 days, enhanced cellular uptake against HCT-116 cells, and higher phototoxicity than Ce6 itself [16]. Polymeric micelles are frequently employed for the site-specific delivery of bioactive agents against tumors [22]. In particular, cyclodextrin, which is a cyclic oligosaccharide, is believed to be an ideal candidate for the solubilization of hydrophobic drugs and site-specific delivery of bioactive agents [20,24,25,26]. Paul et al. reported that hydroxypropyl-β-cyclodextrin increases the aqueous solubility of Ce6 at pH 7 through the formation of an inclusion complex and then improves its phototoxicity against oral squamous carcinoma cells through enhanced ROS production [20]. Li et al. also reported that polydopamine nanoparticles modified with β-cyclodextrin show an excellent drug-loading capacity of doxorubicin and Ce6, resulting in a superior antitumor activity in vitro/in vivo [24]. Nanoplatforms, including β-cyclodextrin, can be passively accumulated in tumor tissues, properly producing singlet oxygen under irradiation and then initiating anti-tumor immune responses against the metastasis of breast cancer cells [25]. Xue et al. reported that β-cyclodextrin-substituted aza–boron–dipyrromethene-based photosensitizers are suitable carriers for the site-specific targeting of cancer cells [26].

In this study, we synthesized succinylated β-cyclodextrin conjugated with Ce6 via a disulfide linkage (bCDsu-ss-Ce6). Redox potentials were elevated in the tumor tissue, and the level of glutathione (GSH), which is a key molecule for the degradation of disulfide bonds, was also elevated in the tumor tissues [27]. bCDsu-ss-Ce6 conjugates may have sensitivity against intracellular GSH levels in cancer cells and then disintegrate intracellularly. The physicochemical and biological properties of bCDsu-ss-Ce6 were investigated in vitro and in vivo.

## 2. Results

### 2.1. Synthesis of bCDsu-ss-Ce6 Conjugates

The synthesis scheme of bCDsu-ss-Ce6 is shown in Figure 1. The carboxylic acid derivatives of bCD were synthesized by conjugation with succinyl anhydride (bCDsu). After that, cystamine was attached to one end of the carboxylic acid to make bCDsu-ss. Finally, Ce6 activated with an EDC/NHS system was conjugated with bCDsu-ss to obtain bCDsu-ss-Ce6. Unreacted substances were removed by the dialysis procedure. To confirm each chemical structure of bCDsu, bCDsu-ss, and bCDsu-ss-Ce6, spectroscopic methods such as NMR, mass spectrometry, and IR were employed.

Ce6 was investigated in terms of its ^1^H and ^13^C NMR spectra, as shown in Appendix A. Appendix A show the ^1^H and ^13^C NMR spectra of bCDsu obtained in D_2_O due to its highly aqueous solubility via succinyl substitution. The directly substituted carbon(C_6_) and neighboring carbon(C_5_) were shifted 3.96 ppm downfield and 1.96 ppm upfield relative to their unsubstituted counterparts (C_6_′, C_5_′), respectively. Also, protons (H_6a_, H_6b_) attached to the directly substituted carbon were shifted downfield by 0.4~0.62 ppm, while protons (H_5_) on the neighboring carbons were shifted downfield by 0.18 ppm. The ^13^C NMR spectra of bCDsu indicated that specific carboxylic acid peaks of bCDsu were observed at 170~180 ppm. The degree of substitution (DS) of the succinyl group could be calculated by comparing the substituted methylene peaks (H_6a_, H_6b_, 2H) to the peaks of anomeric protons (H_1_, 1H), resulting in a value of 6.69. Furthermore, the [M+H]^+^ ion peaks of bCDsu with DS 6 and 7 were observed to be at *m*/*z* 1735.436 and 1835.447, respectively, in the ESI-MS analysis, as depicted in Appendix A.

bCDsu-ss was analyzed in terms of its 1D and 2D NMR spectra, as shown in Appendix A. The ^1^H and ^13^C NMR spectra are shown in Appendix A, respectively. The ^1^H-^13^C HSQC and ^1^H-^13^C HMBC spectra are shown in Appendix A, respectively. The carbons (C_8_, C_9_, C_10_) of the succinyl group that were conjugated with cystamine were differentiated from succinyl carbons (C_8′_, C_9′_, C_10′_) without cystamine at around 30 ppm and 172 ppm. The degree of substitution (DS) of the cystamine group could be calculated by comparing the substituted methylene peaks (H_12_, H_13_, 4H) to the peaks of anomeric protons (H_1_, 7H), resulting in a value of 1.25. Evidently, the correlation peak between the carbonyl carbon (C_10_) of the succinyl group and the methylene protons (H_11_) of the cystamine in the HMBC spectrum indicated a clear conjugation via a peptide bond. And, the ESI-MS spectrum showed a [M+H]^+^ peak at *m*/*z* 1969.520, as shown in Appendix A.

Finally, the bCDsu-ss-Ce6 was confirmed through analyzing its 1D and 2D NMR and IR spectra. As shown in Figure 2A, the ^1^H spectrum revealed a shift of H_15a_ (5.39 ppm) in Ce6 to H_15a_ (5.53 ppm) in bCDsu-ss-Ce6. Figure 2B(a,c) show the intrinsic peaks of the bCDsu-ss and Ce6 in their ^1^H NMR spectra. As shown in Figure 2B(c), the methylene proton (H_15a_, 5.39 ppm in Figure 2B(b)) peak of Ce6 was shifted to 5.53 ppm. In the ^13^C NMR spectrum of Appendix A, most peaks were assigned except for one carbon(C_33_), despite its weak and obscure nature. These peaks were assigned through the projection of the correlation peaks in the HSQC and HMBC spectra. Appendix A show the HSQC and HMBC spectra of Ce6, respectively. Appendix A show the HSQC and HMBC spectra of the bCDsu-ss-Ce6 conjugates. In the HMBC spectra of Ce6 (Appendix A) and bCDsu-ss-Ce6 (Appendix A), significant correlations were observed. In the case of Ce6, the methylene proton (H_15a_, 5.39 ppm) peak showed correlations with α and β position carbons (C_14_, C_15_, C_15b_, C_16_) through two- and three-bond couplings. Similarly, in the HMBC spectrum of bCDsu-ss-Ce6, the formation of an amide bond between Ce6 and cystamine resulted in correlation peaks between the proton (H_15a_, 5.53 ppm) and carbons (C_15_, C_34_, C_16_), providing evidence for the successful synthesis of the bCDsu-ss-Ce6 conjugates. In the FT-IR spectrum of Appendix A, C=O stretching of the amide group of the bCDsu-ss-Ce6 conjugates was observed at 1650 cm^−1^. We could not obtain the MS spectrum of the bCDsu-ss-Ce6 conjugates due to their low solubility and ionization. Appendix A shows the FT-IR spectra of the bCD, bCDsu, bCDsu-ss, Ce6, and bCDsu-ss-Ce6 conjugates. As shown in Appendix A, the intrinsic spectra of bCD and Ce6 were obtained, respectively. When the bCD was reacted with succinic anhydride to produce bCDsu, C=O stretching of the carboxylic acid group was observed at 1720~1740 cm^−1^. When cystamine was attached to the bCDsu to produce bCDsu-ss, C=O stretching of the amide group was observed at around 1650 cm^−1^. As shown in Appendix A, C=O stretching of the amide group of the bCDsu-ss-Ce6 conjugates was also observed at 1650 cm^−1^. These results indicated that bCDsu, bCDsu-ss, and bCDsu-ss-Ce6 conjugates were successfully synthesized.

### 2.2. Characterization of bCDsu-ss-Ce6 Nanophotosensitizers

Nanophotosensitizers of the bCDsu-ss-Ce6 conjugates were fabricated using the dialysis method. Since Ce6 is a hydrophobic molecule, the hydrophobic interaction between the Ce6 in the conjugates was a driving force for forming self-aggregates. These aggregates then formed nanoparticles, as shown in Figure 3a. In other words, the TEM images of bCDsu-ss-Ce6 nanophotosensitizers showed a spherical morphology and a small particle size. Their particle size distribution and zeta potential are shown in Figure 3b,c. As shown in Figure 3b, the bCDsu-ss-Ce6 nanophotosensitizers showed a narrow size distribution. The zeta potential of the bCDsu-ss-Ce6 nanophotosensitizers also showed a narrow size distribution, as shown in Figure 3c.

Table 1 summarizes the characteristics of the bCDsu-ss-Ce6 nanophotosensitizers. As shown in Table 1, the experimental Ce6 contents in the bCDsu-ss-Ce6 nanophotosensitizers were not significantly different compared to the theoretical value of the Ce6 contents, indicating that the Ce6 contents were identical to the characterization from the NMR spectra. The particle size and zeta potential of the bCDsu-ss-Ce6 nanophotosensitizers are shown in Table 1. The particle size was 162.9 ± 24.4 nm as an intensity average. The polydispersity and zeta potential were 0.202 and −10.4 mV.

Figure 4 shows the changes in the fluorescence intensity of the bCDsu-ss-Ce6 nanophotosensitizers according to the GSH concentration and Ce6 release properties in vitro. As shown in Figure 4a, the fluorescence intensity of the bCDsu-ss-Ce6 nanophotosensitizers was gradually increased according to the concentration of GSH, indicating that the disulfide bond could be disintegrated, and then the Ce6 was liberated from the nanophotosensitizers. As shown in Figure 4b, the addition of GSH accelerated the release rate of Ce6 from the nanophotosensitizers, while the Ce6 release was minimized in the absence of GSH. This indicates that the bCDsu-ss-Ce6 nanophotosensitizers exhibited a redox-sensitive Ce6 release behavior in the biological system.

### 2.3. In Vitro Cell Culture Study

Figure 5 shows the intrinsic cytotoxicity of the Ce6 alone, bCDsu-ss, and bCDsu-ss-Ce6 nanophotosensitizers against CCD986Sk cells. All of them were not significantly cytotoxic to the CCD986Sk cells, i.e., the viability of the cells was higher than 80% until 5 µg Ce6/mL of the bCDsu-ss-Ce6 nanophotosensitizers and 50 µg/mL of the bCDsu-ss were used, while Ce6 alone showed that the cell viability was less than 80% at 5 µg Ce6/mL. These results indicated that the bCDsu-ss and bCDsu-ss-Ce6 nanophotosensitizers had no intrinsic cytotoxicity against normal cells.

Figure 6 shows the Ce6 uptake ratio, ROS generation, and phototoxicity of Ce6 alone and of the bCDsu-ss-Ce6 nanophotosensitizers against HeLa cells. As shown in Figure 6a, the Ce6 uptake ratio against the HeLa cells was significantly higher in the treatment with the bCDsu-ss-Ce6 nanophotosensitizers than that of Ce6 alone, indicating that the bCDsu-ss-Ce6 nanophotosensitizers have a higher potential for cellular uptake compared to Ce6. As shown in Figure 6b, the ROS generation gradually increased according to the Ce6 concentration, both for Ce6 alone and for the bCDsu-ss-Ce6 nanophotosensitizers. In particular, the ROS generation of the bCDsu-ss-Ce6 nanophotosensitizers was 2 times higher than that of Ce6 alone, indicating that the bCDsu-ss-Ce6 nanophotosensitizers had a superior efficacy in terms of ROS generation. Figure 6c shows the phototoxicity of Ce6 alone and of the bCDsu-ss-Ce6 nanophotosensitizers in the presence or absence of light irradiation. As shown in Figure 6c, the presence of light irradiation did not significantly affect the viability of the HeLa cells, i.e., the viability of the HeLa cells was higher than 80% until 2 µg Ce6/mL of Ce6 alone and bCDsu-ss-Ce6 nanophotosensitizers were used in the absence of light irradiation. When light was irradiated onto the cells, the cell viability gradually decreased according to the Ce6 concentration. In particular, the phototoxicity of the bCDsu-ss-Ce6 nanophotosensitizers was superior to that of Ce6 alone. In other words, the cell viability at 5 µg Ce6/mL was lower than 20% for the bCDsu-ss-Ce6 nanophotosensitizers, while the treatment with Ce6 alone showed a viability higher than 50% at the same concentration. These results indicated that the bCDsu-ss-Ce6 nanophotosensitizers had a higher efficacy in terms of the generation of ROS and phototoxicity against cancer cells. As summarized in Table 2, the IC_50_ value of the bCDsu-ss-Ce6 nanophotosensitizers with light irradiation was lower than that of Ce6 alone. Specifically, the IC_50_ value of the bCDsu-ss-Ce6 nanophotosensitizers was 1.05 mg/L, while the IC_50_ value of Ce6 was higher than 5 mg/L.

Figure 7 shows the effect of GSH on the Ce6 uptake ratio, ROS generation, and phototoxicity of Ce6 alone or of the bCDsu-ss-Ce6 nanophotosensitizers against the HeLa cells. As shown in Figure 7a,d, a higher GSH concentration induced a higher Ce6 uptake into the cells. As shown in Figure 7d, the addition of GSH after treatment with the bCDsu-ss-Ce6 nanophotosensitizers increased the fluorescence intensity of the HeLa cells, while the addition of GSH did not affect the fluorescence intensity upon treatment with Ce6 alone. Furthermore, the ROS generation also increased according to the increase in the GSH concentration, as shown in Figure 7b. As expected, the phototoxicity also increased according to the increase in the GSH concentration, as shown in Figure 7c. Furthermore, the relative ROS level of the nanophotosensitizers in the absence of GSH was also higher than that of Ce6 itself (Figure 7b). These results indicated that the bCDsu-ss-Ce6 nanophotosensitizers had a redox sensitivity in vitro. As shown in Figure 8, the flow cytometric analysis also supported these results, i.e., the peaks in flow cytometric analysis of the bCDsu-ss-Ce6 nanophotosensitizers were increased compared to Ce6 alone, as shown in Figure 8a,b. Furthermore, the peaks of the cells treated with the bCDsu-ss-Ce6 nanophotosensitizers were increased by the addition of GSH, while Ce6 alone did not significantly change the peaks.

### 2.4. In Vivo Animal Tumor Xenograft Study

Figure 9 shows the biodistribution and PDT efficacy of the nanophotosensitizers against HeLa tumor xenograft models. HeLa cells were administered to the backs of mice to create tumor xenografts in order to study the biodistribution and PDT efficacy of the nanophotosensitizers. As shown in Figure 9a,b, both the Ce6 and bCDsu-ss-Ce6 nanophotosensitizers properly targeted and imaged the tumor mass in the whole body and in each organ image. However, the relative fluorescence intensity of the tumor mass was significantly higher than that of the liver in the nanophotosensitizer treatment, while the changes in the fluorescence intensity were not significant in the Ce6 treatment. These results indicated that the bCDsu-ss-Ce6 nanophotosensitizers have superior potential for targeting tumors. Figure 9c shows the PDT efficacy of the Ce6 and bCDsu-ss-Ce6 nanophotosensitizers against the HeLa tumor xenograft. As shown in Figure 9c, the growth of the tumor mass was inhibited by both the Ce6 and bCDsu-ss-Ce6 nanophotosensitizer treatments under light irradiation. In particular, the tumor mass of the treatment with the bCDsu-ss-Ce6 nanophotosensitizer plus light irradiation was significantly smaller than that of the treatment with Ce6 plus light irradiation. When light irradiation was absent, the growth of the tumor mass was almost similar to PBS for both the Ce6 and bCDsu-ss-Ce6 nanophotosensitizers. These results indicated that the bCDsu-ss-Ce6 nanophotosensitizers have superior potential in terms of tumor targeting and PDT efficacy.

## 3. Discussion

Due to their superior properties, nano-scale carriers such as nanoparticles, polymeric micelles, polymer conjugates, and nanomaterials have been extensively investigated in the field of drug targeting and stimuli-sensitive drug delivery systems [28,29,30,31]. In particular, nanoparticles with small particle sizes have been considered as an appropriate vehicle for hydrophobic anticancer agents due to their potential for the site-specific delivery of bioactive agents, the solubilization of lipophilic agents, and their ease of surface modification for the recognition of specific cells [27,28,29,30,31,32]. For example, Li et al. reported that Anti-HER2 antibody-decorated poly(lactide-co-glycolide)-poly(ethylene glycol) (PLGA-PEG)/ PLGA nanoparticles can be used to target breast cancer cells. In particular, the targeting efficiency can be controlled by the ligand density on the nanoparticle surfaces against breast cancer cells with a controlled release of docetaxel [33]. Mitchell reviewed that nanoparticles can be engineered to have an adjustable particle size, stealth properties via surface modification, and the ability to decorate targeting moieties [21]. These engineered nanoparticles enable us to circumvent biological barriers and target specific cells at the diseased site. Furthermore, polymeric micelles using PEG-polyester block copolymer can be used to improve the solubility of hydrophobic drugs such as coumarin 6 [34]. Supramolecular assembly based on Ce6-α-cyclodextrin with PEGylated-peptide conjugates enhances the Gram-negative bacteria targeting efficiency, and the eradication efficacy of bacterial film is higher than PDT with traditional photosensitizers [34].

In this study, the bCDsu-ss-Ce6 conjugates formed nanoparticles that had small sizes of less than 200 nm, as shown in Figure 3. The driving force of the nanophotosensitizer formation was the hydrophobic interaction between the Ce6 in the bCDsu-ss-Ce6 and its subsequent association in the aqueous solution, which was due to Ce6 being a lipophilic molecule [35,36]. 

Ce6 must be associated by hydrophobic interactions, while bCDsu is a hydrophilic molecule that then forms nano-dimensional carriers. Cyclodextrins have been extensively used in drug delivery and pharmaceutical applications [37,38,39]. Since they have a hydrophobic interior and hydrophilic exterior, cyclodextrins are frequently employed to form complexes with hydrophobic compounds. They are applied to deliver various kinds of drugs and enhance the solubility/stability of these drugs [38]. Furthermore, the inclusion complexes of cyclodextrins with hydrophobic agents are known to improve the penetration efficacy into body tissues and deliver drugs under specific physiological conditions [40]. Paul et al. reported that the formation of an inclusion complex between Ce6 and hydroxypropyl-β-CD (HP-β-CD) induces an increase in the aqueous solubility and disaggregation of Ce6 [20]. They argued that Ce6 alone shows a low singlet oxygen (SO) generation efficiency because Ce6 molecules are largely aggregated in the aqueous solution. These inclusion complexes improve PDT’s efficacy against oral squamous carcinoma cells through enhanced singlet oxygen (SO) generation and phototoxicity. HP-β-CD induces the disaggregation of Ce6 into an aqueous solution, and then these inclusion complexes improve SO generation in the aqueous solution. Li et al. also reported that orthogonal assemblies between polydopamine-modified β-CD improved the synergistic anticancer effect [24]. In our results, the bCDsu-ss-Ce6 nanophotosensitizers showed a higher intracellular uptake and PDT efficacy, as shown in Figure 9. The mechanisms of the intracellular uptake of the nanophotosensitizers were practically governed by non-specific endocytosis or absorptive endocytosis because the bCDsu-ss-Ce6 nanophotosensitizers have no targeting moiety on the nanoparticle surface [41]. Thakur et al. also reported that lipid–polymer hybrid nanoparticles encapsulating zinc-phthalocyanine and quercetin have a synergistic effect on the intracellular uptake and PDT against breast cancer cells [42].

The tumor microenvironment is known to have quite a different physiological status compared to normal tissues, i.e., the tumor microenvironment is characterized by an acidic pH, an enhanced permeation/retention (EPR) effect of molecules, abundant enzymes, over-expressed molecular receptors, and an abnormal redox potential [43,44,45,46]. In particular, the abnormal redox status of the tumor microenvironment is known to increase the GSH level in cancer cells and then induce drug-resistant problems with anticancer agents following chemotherapy failure [47,48]. Our results also indicated that the bCDsu-ss-Ce6 nanophotosensitizers efficiently accumulated Ce6 intracellularly and generated ROS levels that were two times higher than Ce6 alone (Figure 6a,b). These results improved cancer cell death in vitro and in vivo, as shown in Figure 6c and Figure 9. The bCDsu-ss-Ce6 nanophotosensitizers could be delivered intracellularly into HeLa cells in a higher redox status, i.e., the Ce6 uptake ratio was significantly higher due to the addition of GSH, which then increased the ROS generation and phototoxicity, as shown in Figure 7. These results must be due to the fact that Ce6 could be liberated from the nanophotosensitizers in the elevated redox status, as shown in Figure 6. Thus, our bCDsu-ss-Ce6 nanophotosensitizers have a redox-sensitive potential. In addition, Parkhats et al. reported that the quantum yield of SO production of Ce6 in an aqueous solution was significantly decreased at acidic pH values compared to basic pH values, while the quantum yield of polyvinylpyrrolidone (PVP)–Ce6 conjugates at acidic pH values was not significantly decreased [49]. They argued that the Ce6 itself was aggregated in the acidic pH conditions, and then this phenomenon induced a lower quantum yield at lower pH values. However, the Ce6 aggregation could be inhibited in the PVP–Ce6 conjugates, after which they maintained a higher quantum yield. Furthermore, the Ce6 accumulation, ROS production, and phototoxicity were increased according to the increase in the GSH concentration, as shown in Figure 7. A higher GSH concentration induced a stronger red fluorescence intracellular intensity, as shown in Figure 7d. These results might be due to the fact that the intracellular uptake of the nanophotosensitizers occurred dominantly.

Ce6 was liberated in the intracellular GSH molecules, and after that, the fluorescence intensity increased, as illustrated in Figure 10. Also, Ce6 might already have been liberated from the nanophotosensitizers in the extracellular GSH molecules, and then the liberated Ce6 could have entered intracellularly. Otherwise, the addition of GSH only resulted in a small increase in ROS generation, as shown in Figure 7b. These results might be due to the reducing effect of GSH, which then competes with the ROS generation. Defensive mechanisms in cancer cells may affect the efficacy of PDT, which means that ROS generation can be controlled by intracellular GSH levels [50]. The effect of the intracellular GSH level on the ROS generation and the PDT effect will be discussed in the next report. The bCDsu-ss-Ce6 nanophotosensitizers efficiently accumulated in the tumor tissues. These phenomena also resulted in an enhanced anticancer activity of the bCDsu-ss-Ce6 nanophotosensitizers, suggesting that the bCDsu-ss-Ce6 nanophotosensitizers are suitable candidates for theranostic cancer treatment.

## 4. Materials and Methods

### 4.1. Materials

β-cyclodextrin (bCD) was purchased from Combi-Blocks, Inc. (San Diego CA92126, USA). From the manufacturer’s information, the M.W. of bCD is 1135 g/mol. Succinic anhydride, N-(3-dimethylaminopropyl)-N′-ethylcarbodiimide hydrochloride (EDC), N-hydroxysuccinimide (NHS), cystamine dihydrochloride, triethylamine (TEA), L-glutathione (GSH), pyridine, dimethyl sulfoxide (DMSO), dimethyl sulfoxide (DMSO-D6), 2,2,2-tribromoethanol (avertin), and deuterium oxide (D_2_O) were purchased from Sigma Aldrich Co. Ltd. (St. Louis, MO, USA). Chlorin e6 (Ce6) was obtained from Frontier Scientific Co. (Logan, UT, USA). All chemicals and reagents were used without further purification or treatment. Dialysis membranes with molecular weight cut-offs (MWCO) of 1000 g/mol and 2000 g/mol were obtained from Spectra/Por^TM^ Membranes (Rancho Dominguez, CA, USA). Avertin (2,2,2-tribromoethanol) and tert-amyl alcohol were purchased from Sigma Aldrich Chem. Co. (St. Louis, MO, USA). All experiments were performed at room temperature.

### 4.2. Instruments

Synthesis of bCDsu and bCDsu-ss-Ce6 was confirmed with nuclear magnetic resonance (NMR) spectroscopy [28]. Samples dissolved in D_2_O or DMSO-d6 were used to analyze the products with NMR spectroscopy (Agilent VNMRS 600 MHz spectrometer with liquid-helium-cooled cryoprobe, Santa Clara, CA, USA). Spectra were obtained at 25 °C and processed with the MestreNova (Mnova) software, version 12.0.0.

Mass spectra were acquired on a quadrupole time-of-flight mass spectrometer (Xevo G2-XS, Waters, Cambridge, UK) equipped with an electrospray ionization source at Chonnam National University.

Particle size distribution was measured with a Zetasizer (Nano-ZS, Malvern, Worcestershire, UK). The effect of GSH on the particle size was investigated as follows: GSH was added to an aqueous solution of bCDsu-ss-Ce6 nanophotosensitizers in phosphate-buffered saline (PBS, pH 7.4, 0.01 M) and then incubated at 37 °C for 3 h. This solution was used to measure the particle size.

The morphology of the nanophotosensitizers was observed with transmission electron microscopy (TEM) (H7600, Hitachi Instruments Ltd., Tokyo, Japan). One drop of an aqueous solution of the nanophotosensitizers was placed onto a carbon-film-coated grid, and then this was dried at room temperature. For negative staining, phosphotungstic acid (0.1%, *w*/*w* in H_2_O) was used. TEM observation was carried out at 80 kV.

### 4.3. Synthesis of bCDsu-ss-Ce6 Conjugates

bCD was reacted with succinic anhydride to synthesize succinylated bCD (bCDsu), as reported previously [51]. Ce6 was conjugated with bCDsu via cystamine linkage. ThebCDsu-ss-Ce6 conjugates were synthesized by the conjugation of succinylated bCD with Ce6 via cystamine linkage.

***Succinylated β-cyclodextrin (bCDsu)***: bCD (2.27 g, 2 mmol) was dissolved in 15 mL of pyridine, and an excess quantity of succinic anhydride (2.8 g, 28 mmol) was added to the solution and stirred for 16 h at room temperature under a nitrogen atmosphere. The pyridine was then evaporated in a smart evaporator under vacuum at 50 °C and then acetone was added to the precipitate. The acetone mixture was filtrated and washed with acetone 2~3 times, and resulting precipitate was dried overnight. The resultant products weighed 3.13 g, and the yield was 85% (*w*/*w*).

***Cystamine-conjugated bCDsu (bCDsu-ss)***: bCDsu (300 mg, 0.163 mmol) dissolved in DMSO (15 mL) was mixed with 4 equivalents each of EDC (125.33 mg,0.654 mmol) and NHS (75.25 mg, 0.654 mmol). This solution was stirred for 3 h at room temperature under a nitrogen atmosphere. After that, cystamine dihydrochloride (92.02 mg, 0.409 mmol) and TEA (41.35 mg, 0.409 mmol) dissolved in DMSO (5 mL) were added dropwise into the stirred solution. The reaction mixture was magnetically stirred for 4 h at room temperature under a nitrogen atmosphere. The resulting solution was introduced into a dialysis membrane (MWCO: 1000 g/mol) and then dialyzed against deionized water for 2 days. The deionized water was exchanged every 3 h for 2 days. After that, the dialyzed solution was freeze-dried for 2 days, and bCDsu-ss was obtained as a white solid. The resultant products weighed 178 mg, and the yield was 52% (*w*/*w*).

***Ce6-**conjugated bCDsu-ss (bCDsu-ss-Ce6)***: Ce6 (21.82 mg, 0.0366 mmol) dissolved in DMSO (6 mL) was mixed with 1.5 equivalents each of EDC (10.52 mg, 0.0548 mmol) and NHS (6.312 mg, 0.0548 mmol). This solution was stirred for 4 h at room temperature under a nitrogen atmosphere. bCDsu-ss (100 mg, 0.0475 mmol) dissolved in DMSO (4 mL) was added dropwise into the stirred solution. The reaction mixture was magnetically stirred for 2 days at room temperature under a nitrogen atmosphere. The resulting solution was introduced into a dialysis membrane (MWCO: 1000 Da) and then dialyzed against DMSO for a few hours and then against deionized water for 2 days. The deionized water was exchanged every 3 h. The dialyzed solution was freeze-dried for 2 days, and bCDsu-ss-Ce6 was obtained as a dark solid. The resultant products weighed 55 mg, and the yield was 59% (*w*/*w*).

### 4.4. Fabrication of bCDsu-ss-Ce6 Nanophotosensitizers

bCDsu-ss-Ce6 (20 mg) was reconstituted in 2 mL of deionized water, and then DMSO (3 mL) was added. This solution was poured into 10 mL of deionized water and introduced into a dialysis membrane (MWCO: 2000 Da). This solution was then dialyzed against deionized water for 1 day with the deionized water being exchanged at 3 h intervals. The resulting solution was either used for analysis or lyophilized for 2 d.

### 4.5. Fluorescence Spectra

Fluorescence emission spectra of the nanophotosensitizers were measured with a fluorescence spectrofluorophotometer (Shimadzu RF-5301PC spectrofluorophometer, Kyoto, Japan). The Ce6 concentration of the nanophotosensitizers was adjusted to 0.1 mg/mL in PBS with or without GSH. This solution was allowed to stand for 3 h at 37 °C, and then the fluorescence emission spectra were recorded between 600 nm and 800 nm (excitation wavelength: 400 nm). The fluorescence images were acquired with a Maestro 2 small-animal-imaging instrument (Cambridge Research and Instrumentation Inc., Hopkinton, MA, USA).

### 4.6. Drug Release Study

The nanophotosensitizers were prepared as described in Section 2.4. The volume of the nanophotosensitizer solution was adjusted to 20 mL (1 mg/mL as bCDsu-ss-Ce6 conjugates) with deionized water. This solution (5 mL) was introduced into a dialysis membrane (MWCO: 2000 Da) and then transferred into 45 mL of PBS (pH 7.4, 0.01 M) in a conical tube. GSH was added to this solution to study the effect of GSH on the drug release properties. These solutions were incubated with a shaking incubator (100 rpm) at 37 °C. The external PBS solution was collected at predetermined time intervals to measure the liberated Ce6, and then fresh PBS was added to the vials. For measurement of the Ce6 concentration, collected samples were measured with a UV-VIS spectrophotometer (Genesys 10s UV-VIS spectrophotometer, Thermo Fisher Scientific, Waltham, MA, USA) at 664 nm. The results were expressed as the average ± standard deviation (S.D.) from three separated experiments.

To measure the experimental drug contents, 5 mg of the bCDsu-ss-Ce6 nanophotosensitizers were reconstituted in 20 mL of PBS with 20 mM GSH and then incubated at 37 °C for one day with a shaking incubator (100 rpm). This solution was diluted with DMSO more than 10 times and then measured with a UV spectrophotometer. Drug contents = (Ce6 weight/nanophotosensitizer weight) × 100. For comparison, similar weights of bCDsu-ss and GSH were used as blanks.

### 4.7. Cell Culture Study

***Cells***: CCD986sk human skin fibroblast cells and HeLa human cervical cancer cells were obtained from Korean Cell Line Bank, Co. (Seoul, Republic of Korea). CCD986sk cells were cultured using an IMDM (Gibco, Grand Island, NY, USA) medium supplemented with 10% heat-inactivated fetal bovine serum (FBS) (Invitrogen) and 1% penicillin/streptomycin. HeLa cells were cultured using an MEM (Gibco, Grand Island, NY, USA) medium supplemented with 10% heat-inactivated fetal bovine serum (FBS) (Invitrogen) and 1% penicillin/streptomycin. All cell lines were maintained at 37 °C in a 5% CO_2_ incubator.

***Intracellular Ce6 uptake***: HeLa cells that were seeded into 96-well plates (2 × 10^4^ cells/well) were exposed to either Ce6 or the bCDsu-ss-Ce6 nanophotosensitizers and, 2 h later, the cells were washed with PBS twice. For the Ce6 treatment, Ce6 dissolved in DMSO (1 mg Ce6/mL DMSO) was diluted with serum-free media more than 200 times (DMSO final concentration: 0.5% (*v*/*v*)) to make a Ce6 concentration of 5 µg/mL and then diluted with serum-free media to an appropriate concentration. The nanophotosensitizers in deionized water were filtered with a syringe filter (1.2 µm) and then diluted with serum-free media more than 10 times. The cells were lysed with 50 µL of lysis buffer (GenDEPOT, Barker, TX, USA) to analyze the Ce6 contents in the cells. The intracellular Ce6 uptake ratio was analyzed with the relative fluorescence intensity with an Infinite M200pro microplate reader (Tecan, Mannedorf, Switzerland) (excitation wavelength: 407 nm, emission wavelength: 664 nm).

Fluorescence microscopy was used to observe the cell morphology. Cells seeded on the cover glass in six-well plates (2 × 10^5^ cells/well) were treated with either Ce6 or the bCDsu-ss-Ce6 nanophotosensitizers (Ce6 concentration: 2 µg/mL). After 1 h, the cells were washed with PBS twice, and then the cells were fixed with 4% paraformaldehyde (PFA) solution in PBS for 15 m. These were immobilized with immobilization solution (Immunomount, Thermo Electron Co. Pittsburgh, PA, USA) and observed with a fluorescence microscope (Eclipse 80i, Nikon, Tokyo, Japan).

Flow cytometry analysis of the HeLa cells was performed using a flow cytometer (NAVIOS, Beckman Coulter Inc., Brea, CA, USA). The cells (2 × 10^5^ cells) were exposed to either Ce6 or the bCDsu-ss-Ce6 nanophotosensitizers (Ce6 concentration: 2 µg/mL). After 1 h, the cells were washed with PBS twice and then adopted to measure them with a flow cytometer using a red (635 nm) laser filter at room temperature.

***Intracellular ROS generation***: The intracellular ROS generation of the HeLa cells was analyzed with DCFH-DA. The cells (2 × 10^4^ cells/well), which were seeded in a 96-well plate, were exposed to either Ce6 or the bCDsu-ss-Ce6 nanophotosensitizers in serum-free media, and DCFH-DA in PBS was also added (final concentration: 20 µM). After 2 h, the cells were washed with PBS twice, replaced with fresh phenol-red-free RPMI media (100 µL), and then irradiated at 664 nm (2.0 J/cm^2^) using an expanded homogenous beam (SH Systems, Gwangju, Korea). The intracellular ROS level was analyzed with a microplate reader (Infinite M200 PRO, Tecan, Mannedorf, Switzerland) (excitation wavelength, 485 nm; emission wavelength, 535 nm). All procedures were performed in dark conditions.

***PDT study against cancer cells***: HeLa cells that were seeded in 96-well plates (2 × 10^4^ cells/well) were exposed to either Ce6 alone or the bCDsu-ss-Ce6 nanophotosensitizers in a Ce6 concentration range of 0.001~5 µg/mL. The cells were incubated for 2 h in a 5% CO_2_ incubator at 37 °C and then washed with PBS twice. To these cells, 100 µL of serum-free media was added, and the cells were then irradiated at 664 nm (light dose: 2.0 J/cm^2^) using an expanded homogenous beam (SH Systems, Gwangju, Korea). After that, the cells were incubated for 24 h in a 5% CO_2_ incubator at 37 °C. Cell viability was analyzed with an MTT assay. MTT solution (30 µL from MTT stock solution (5 mg/mL in PBS)) was added to the 96 wells, which were then placed into a CO_2_ incubator for 4 h. After that, the supernatants were discarded and then replaced with 100 µL of DMSO. The viability of the cells was analyzed by absorbance at 570 nm using an Infinite M200 PRO microplate reader. All procedures for the PDT study were carried out in dark conditions.

An intrinsic dark toxicity of the CCD986Sk cells and HeLa cells was performed with the same procedure in the absence of light irradiation in dark conditions.

### 4.8. Animal Tumor Imaging Using HeLa Tumor Xenograft Model In Vivo

For the animal study, nude BALb/C mice (male, 20 g, five weeks old, OrientBio Co. Ltd. Seongnam, Korea) were used. HeLa cells (1 × 10^6^ cells) were subcutaneously (s.c.) administered into the backs of nude BALb/C mice. When the diameter of the tumor xenograft became bigger than 6 mm, the HeLa-cell-bearing mice were used for fluorescence imaging. Either an aqueous Ce6 solution or a nanophotosensitizer solution (10 mg Ce6/kg) was intravenously (i.v.) injected via the tail veins of the mice (injection volume: 200 µL). For the Ce6 treatment, Ce6 was dissolved in ethanol/Cremophor EL solution (1/1) and then diluted with PBS more than 10 times. The nanophotosensitizers in deionized water were filtered with a syringe filter (1.2 µm). A Maestro^TM^ 2 small-animal-imaging instrument (Cambridge Research and Instruments, Inc., Woburn, MA, USA) was used to observe the fluorescence images of the whole bodies of the mice. For imaging the animals, the mice were anesthetized with avertin. For the biodistribution of either Ce6 or the nanophotosensitizers, the organs of the mice were extracted. To anesthetize the animals, 0.5 mL of a stock solution of avertin (25 g of avertin in 15.5 mL of tert-amyl alcohol) was mixed with 39.5 mL of 0.9% saline solution. The avertin solution (300~400 µL/mice) was intraperitoneally (i.p.) administered to anesthetize the mice.

### 4.9. PDT of HeLa Tumor Xenograft Model In Vivo

The PDT of the mice was performed as follows: HeLa cells (1 × 10^6^ cells) were subcutaneously (s.c.) administered into the backs of nude BALb/C mice, and the mice were divided into three groups: control group, Ce6 treatment group, and nanophotosensitizer treatment group. Each group was composed of 5 mice. When the tumor diameters became about 4 mm, either Ce6 alone or the nanophotosensitizers were i.v. administered (injection dose: 10 mg Ce6/kg for each mouse). For the control treatment, PBS was administered. Each treatment was administered intravenously via a tail vein of each mouse. The injection volume was 200 µL. After 2 days of administration, the mice were anesthetized with avertin for PDT (2.0 J/cm^2^, 664 nm). To irradiate the tumor xenograft of the mice, the mice were covered with fabric material to avoid interference from irradiated light, except for the tumor. Day 0 was determined as the day of the first irradiation. Three days later, the mice were irradiated once more. The growth of the tumor volume was measured with vernier calipers at 5-day intervals. The tumor volume was calculated as follows: tumor volume (mm^3^) = (length × width^2^)/2.

The animal experiments were carefully performed under the guidelines of the Pusan National University Institutional Animal Care and Use Committee (PNUIACUC). The protocols of this study were reviewed and monitored by the PNUIACUC in accordance with their ethical procedures and scientific care (approval number: PNU-2020-2751).

### 4.10. Statistical Analysis

The statistical significance was estimated with Student’s *t*-test using SigmaPlot^®^ (SigmaPlot^®^ v.11.0, Systat Software, Inc., San Jose, CA, USA). The minimal level of significance was evaluated as *p* < 0.05.

## 5. Conclusions

bCDsu-ss-Ce6 conjugates were synthesized for the redox-sensitive delivery of a photosensitizer, Ce6, and for studying their PDT efficacy against HeLa. bCD was succinylated to endow carboxyl groups on the bCD (bCDsu), and then Ce6 was conjugated with bCDsu via a disulfide linkage (bCDsu-ss-Ce6). The chemical composition of the bCDsu-ss-Ce6 was confirmed with ^1^H and ^13^C NMR spectra, indicating that Ce6 was conjugated with bCDsu via a disulfide linkage. The bCDsu-ss-Ce6 conjugates formed nano-scale particles in an aqueous solution, i.e., the bCDsu-ss-Ce6 nanophotosensitizers have small particles of less than 200 nm with an average particle size of 152.0 ± 23.2 nm. The Ce6 release rate from the bCDsu-ss-Ce6 nanophotosensitizers was accelerated according to the GSH concentration, indicating that the bCDsu-ss-Ce6 nanophotosensitizers have redox-sensitive drug release properties. The bCDsu-ss-Ce6 nanophotosensitizers have a low intrinsic cytotoxicity against CCD986Sk cells, as does Ce6 alone. The bCDsu-ss-Ce6 nanophotosensitizers showed an enhanced Ce6 uptake ratio, higher ROS production, and improved phototoxicity compared to Ce6 alone. Furthermore, the bCDsu-ss-Ce6 nanophotosensitizers showed a higher Ce6 uptake ratio, ROS generation, and phototoxicity against HeLa cells in the presence of GSH. This indicates that the bCDsu-ss-Ce6 nanophotosensitizers have a redox-sensitive biological activity in an in vitro cell culture system. The bCDsu-ss-Ce6 nanophotosensitizers efficiently accumulated in the tumor tissue, i.e., their fluorescence intensity was significantly higher in the tumor tissues than in other organs compared to Ce6 alone. These results indicated a higher anti-cancer activity of the bCDsu-ss-Ce6 nanophotosensitizers, i.e., they efficiently inhibited the growth of tumors in an in vivo animal tumor xenograft study.

## Figures and Tables

**Figure 1 molecules-28-07398-f001:**
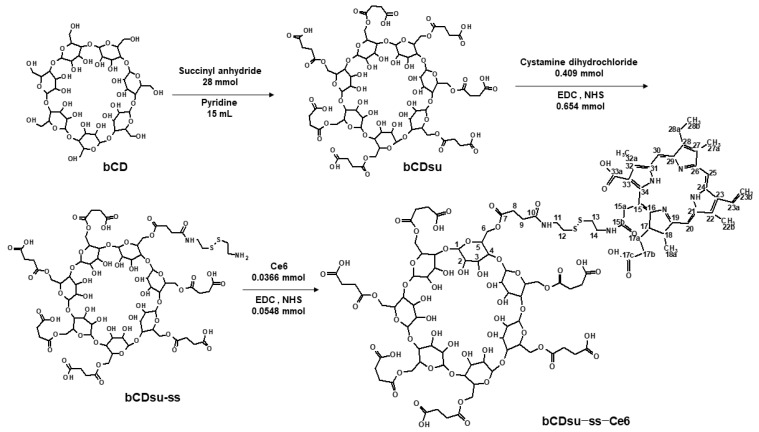
Synthesis scheme of bCDsu-ss-Ce6 conjugates.

**Figure 2 molecules-28-07398-f002:**
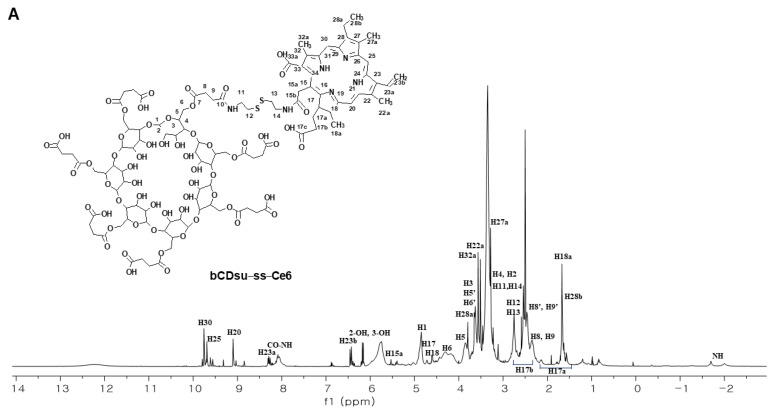
(**A**) ^1^H NMR spectrum assignment of bCDsu-ss-Ce6 conjugates. (**B**) ^1^H NMR spectra of (**a**) bCDsu-ss, (**b**) Ce6, and (**c**) bCDsu-ss-Ce6.

**Figure 3 molecules-28-07398-f003:**
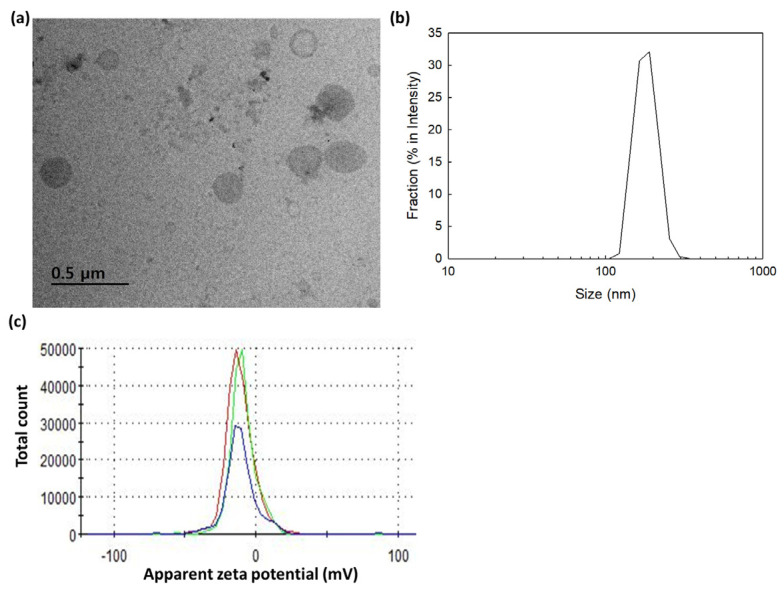
Characterization of bCDsu-ss-Ce6 nanophotosensitizers. (**a**) TEM photograph; (**b**) typical particle size distribution; (**c**) zeta potential.

**Figure 4 molecules-28-07398-f004:**
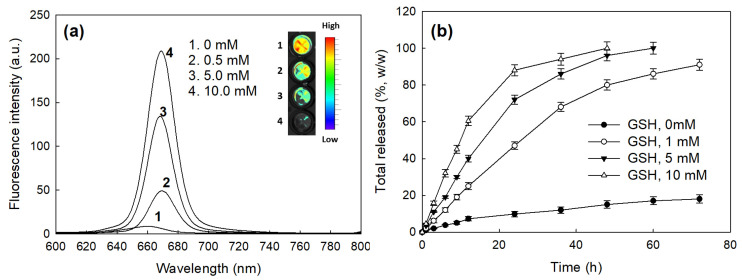
(**a**) The effect of GSH addition on the changes in fluorescence intensity of bCDsu-ss-Ce6 nanophotosensitizers. (**b**) The effect of GSH addition on the changes in Ce6 release from bCDsu-ss-Ce6 nanophotosensitizers.

**Figure 5 molecules-28-07398-f005:**
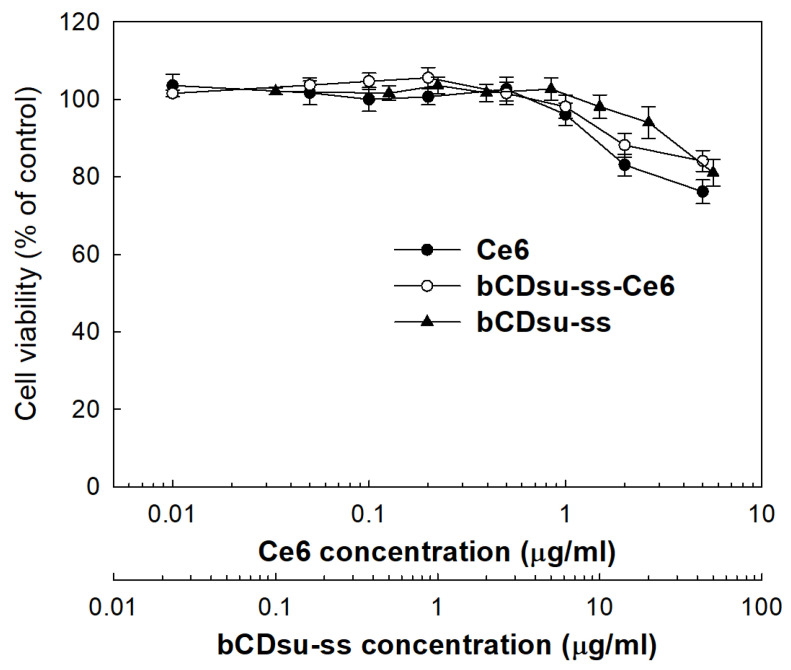
Intrinsic cytotoxicity of Ce6 alone, bCDsu-ss, and bCDsu-ss-Ce6 nanophotosensitizers against CCD986Sk cells. CCD986Sk cells were exposed to Ce6 or nanophotosensitizers in dark conditions.

**Figure 6 molecules-28-07398-f006:**
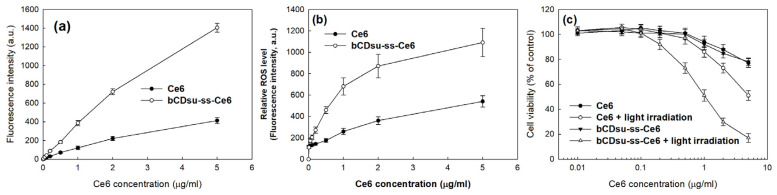
(**a**) Ce6 uptake ratio of Ce6 alone or bCDsu-ss-Ce6 nanophotosensitizers. (**b**) ROS generation of Ce6 alone or bCDsu-ss-Ce6 nanophotosensitizers under light irradiation. (**c**) PDT efficacy of Ce6 alone or bCDsu-ss-Ce6 nanophotosensitizers against HeLa cells in the absence or presence of light irradiation.

**Figure 7 molecules-28-07398-f007:**
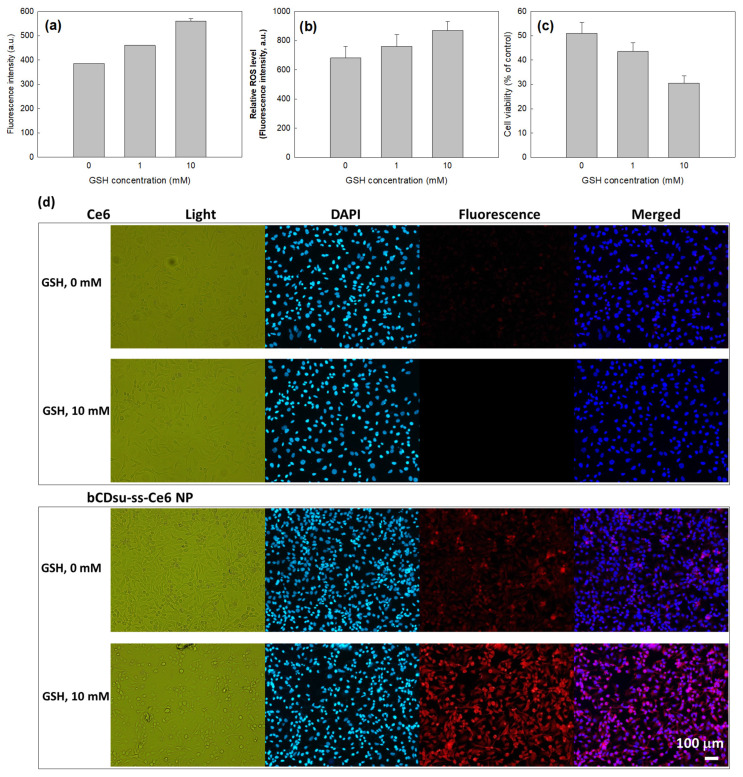
The effect of GSH addition on the HeLa cell culture. (**a**) Ce6 uptake ratio; (**b**) ROS generation; (**c**) phototoxicity of bCDsu-ss-Ce6 nanophotosensitizers under light irradiation; (**d**) the effect of GSH addition on the intracellular uptake of Ce6 or bCDsu-ss-Ce6 nanophotosensitizers. Bar = 100 µm.

**Figure 8 molecules-28-07398-f008:**
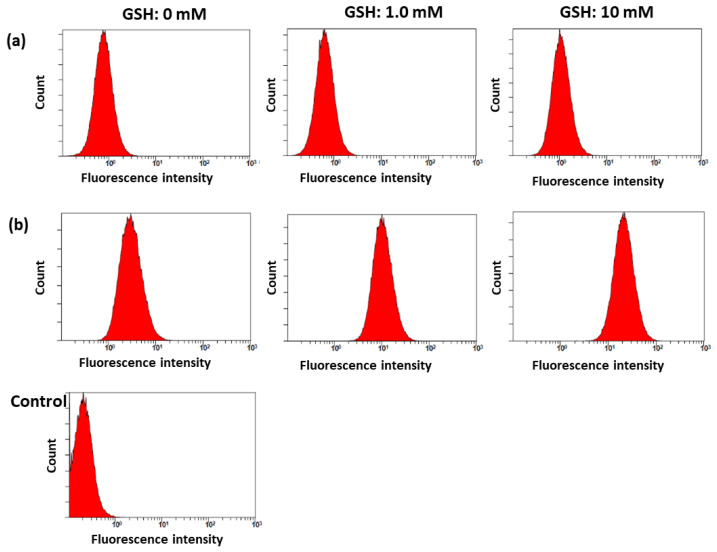
Flow cytometric analysis of HeLa cell culture. Cells were treated with Ce6 alone (**a**) and bCDsu-ss-Ce6 nanophotosensitizers (**b**).

**Figure 9 molecules-28-07398-f009:**
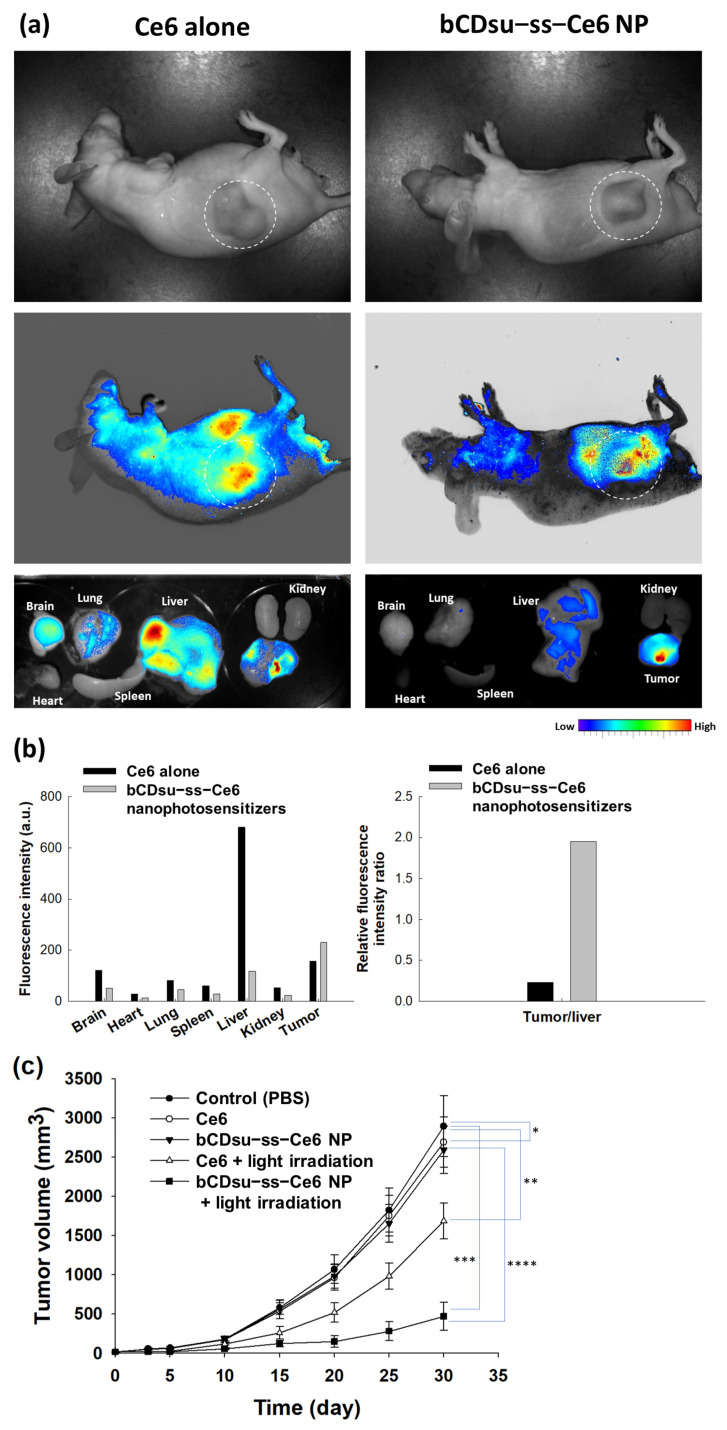
(**a**) In vivo animal imaging using a HeLa-cell-bearing tumor xenograft mouse model. (**b**) Biodistribution of Ce6 alone or of bCDsu-ss-Ce6 nanophotosensitizers (values derived from (**a**)). (**c**) PDT efficacy of Ce6 alone or of bCDsu-ss-Ce6 nanophotosensitizers (bCDsu-ss-Ce6 nanophotosensitizers, NP) with or without light irradiation on the tumor growth (Ce6 dose = 10 mg kg^−1^). *, **: *p* < 0.01; ***, ****: *p* < 0.01.

**Figure 10 molecules-28-07398-f010:**
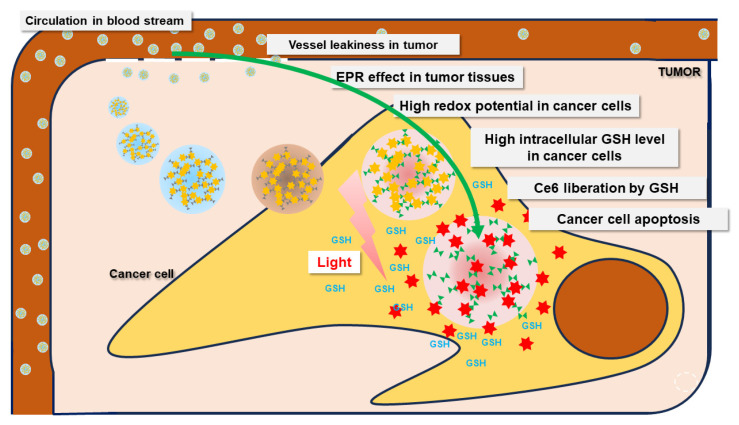
Redox-sensitive delivery of bCDsu-ss-Ce6 against the tumor microenvironment.

**Table 1 molecules-28-07398-t001:** Characterization of bCDsu-ss-Ce6 nanophotosensitizers.

Ce6 Contents (%, *w*/*w*) ^a^		Particle Size Distribution (nm)			Polydispersity	Zeta Potential (mV)
NMR	UV	Int. Ave.	Vol. Ave.	Num. Ave.		
17.23	17.3	162.9 ± 24.4	152.0 ± 23.2	142.7 ± 20.2	0.202	−10.4

^a^ Experimental drug contents were calculated from measurements from NMR and UV spectrophotometer. Drug contents = (Ce6 weight/nanophotosensitizer weight) × 100.

**Table 2 molecules-28-07398-t002:** IC_50_ of Ce6 and bCDsu-ss-Ce6 nanophotosensitizers against HeLa cells.

	IC_50_ (mg/L) ^1^
Ce6	n.d. ^2^
Ce6 + light irradiation	5.16
bCDsu-ss-Ce6 NP	n.d.
bCDsu-ss-Ce6 NP + light irradiation	1.05

^1^ IC_50_ values were estimated from Figure 6b. ^2^ n.d. = not determined.

## Data Availability

Not applicable.

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
