# Peer review of "β-Cyclodextrin Nanophotosensitizers for Redox-Sensitive Delivery of Chlorin e6"

_molecules, 2023, doi:10.3390/molecules28217398_

Round 1

Reviewer 1 Report

Comments and Suggestions for Authors

The manuscript “β-cyclodextrin nanophotosensitizers for redox-sensitive delivery of chlorin e6” is original and actual study written in a clear form. The work considers the synthesis and properties of cyclodextrin-chlorin conjugate. This study brings a contribution in field of anticancer drugs chemistry. The conclusions are useful for medical researchers working in area of photosensitizers and treatment for cancer.

Some paragraphs must be improved for better clarity:

1. The experimental section declares “bCD was reacted with succinic anhydride as reported previously [54]”. But the reference 54 contains no reaction with succinic anhydride.

2. The experimental section mentions “bCD (2.27 g, 2 mmol)”, thus, molar weight is 1135 what corresponds to anhydrous bCD. It is well known that bCD is sold by Sigma Aldrich as a hydrate (~10% wt. water), not anhydrous form. With a previous statement “All chemicals and reagents were used without further purification or treatment” it can mean an experimental error: a hydrate was calculated as an anhydrous form. Such issue must be thoroughly checked.

3. English leaves much to be desired. I recommend to edit the text with some additional help. E.g. Line 90: “showed” must be “shown”.

4. Paragraphs inside Section 4 have wrong numbering (2.x).

5. Compounds purchased from Sigma Aldrich must be declared with corresponding Product Number for a better clearance (since Sigma Aldrich provides a large scale of purity, preparation methods and manufacturers).

Comments on the Quality of English Language

Line 90: “showed” must be “shown”. In many cases, the verb forms must be improved.

Author Response

The manuscript “β-cyclodextrin nanophotosensitizers for redox-sensitive delivery of chlorin e6” is original and actual study written in a clear form. The work considers the synthesis and properties of cyclodextrin-chlorin conjugate. This study brings a contribution in field of anticancer drugs chemistry. The conclusions are useful for medical researchers working in area of photosensitizers and treatment for cancer.

Some paragraphs must be improved for better clarity:

  1. The experimental section declares “bCD was reacted with succinic anhydride as reported previously [54]”. But the reference 54 contains no reaction with succinic anhydride.

Answer) Thanks for your valuable comment. According to your comment, we changed the reference [54].

  1. Vijay V.; Seunho J. Succinyl-β-cyclodextrin–driven synthesis of a nitrogen-fused five-ring heterocycle using GBB-based [4 + 1] cycloaddition via supramolecular host–guest interactions. Tetrahedron. 2019, 75, 778-783

  1. The experimental section mentions “bCD (2.27 g, 2 mmol)”, thus, molar weight is 1135 what corresponds to anhydrous bCD. It is well known that bCD is sold by Sigma Aldrich as a hydrate (~10% wt. water), not anhydrous form. With a previous statement “All chemicals and reagents were used without further purification or treatment” it can mean an experimental error: a hydrate was calculated as an anhydrous form. Such issue must be thoroughly checked.

Answer) Thanks for your valuable comment. I apologize I and my colleagues did some mistake in the section of “4.1. Materials”. Practically, we used beta-cyclodextrin from Combi-Blocks. Inc. USA. We corrected the source of bCD in the 4.1. Materials section. Thanks very much. We considered the purity of bCD (from Combi-Blocks), calculated the molecular weight and weighted bCD for synthesis.

4.1. Materials

β-cyclodextrin (bCD) was purchased from Combi-Blocks, Inc. (San Diego CA92126, USA). From manufacturer’s information, M.W. of bCD is 1135 g/mol. Succinic anhydride, N-(3-Dimethylaminopropyl)-N′-ethylcarbodiimide hydrochloride (EDC), N-hydroxysuccinimide (NHS), cystamine dihydrochloride, triethylamine (TEA), L-glutathione (GSH), pyridine, dimethyl sulfoxide (DMSO), dimethyl sulfoxide (DMSO-D6), 2,2,2-tribromoethanol (avertin) and deuterium oxide(D2O) were purchased from Sigma Aldrich Co. Ltd. (St. Louis, MO, USA). Chlorin e6 (Ce6) was obtained from Frontier Scientific Co. (Logan, UT, USA). All chemicals and reagents were used without further purification or treatment. The dialysis membranes having molecular weight cut-offs (MWCO) of 1000 g/mol and 2000 g/mol were obtained from Spectra/PorTM Membranes. (Rancho Dominguez, CA, USA). Avertin (2,2,2-tribromoethanol) and tert-amyl alcohol were purchased from Sigma Aldrich Chem. Co. (St. Louis, MO, USA). All experiments were performed in room temperature.

  1. English leaves much to be desired. I recommend to edit the text with some additional help. E.g. Line 90: “showed” must be “shown”.

Answer) Thanks for your valuable comment. According to your comment, we corrected “Shoed” to “Shown”. Additionally, we checked grammatical error with aid of native speaker. Thanks again.

  1. Paragraphs inside Section 4 have wrong numbering (2.x).

Answer) Thanks for your valuable comment. According to your comment, we corrected this. Thanks.

  1. Compounds purchased from Sigma Aldrich must be declared with corresponding Product Number for a better clearance (since Sigma Aldrich provides a large scale of purity, preparation methods and manufacturers).

Answer) Thanks for your valuable comment. We checked all the compounds, which are purchased from Sigma as follows:

Succinic anhydride: 8006830100

N-(3-Dimethylaminopropyl)-N′-ethylcarbodiimide hydrochloride (EDC): 8009070001

N-hydroxysuccinimide (NHS): 56480-100G

cystamine dihydrochloride: 30050-100G-F

triethylamine (TEA): 90340-4X25ML

L-glutathione (GSH): G6529-5G

Pyridine: 270407-100ML

dimethyl sulfoxide (DMSO): D8418-500ML

dimethyl sulfoxide (DMSO-D6): 296147-50G

2,2,2-tribromoethanol (avertin): T48402-25G

deuterium oxide(D2O): 151882-100G

tert-amyl alcohol: 8061931000

Comments on the Quality of English Language

Line 90: “showed” must be “shown”. In many cases, the verb forms must be improved.

Answer) Thanks for your valuable comment. According to your comment, we checked the manuscript and corrected the grammatical error.

Reviewer 2 Report

Comments and Suggestions for Authors

This manuscript by Jeong et al reports the preparation of CD-SS-Ce6 conjugate, and the photophysical properties and applications after self-assembling into nanoparticles. These nanoparticles showed redox-responsive function and enhanced therapeutic efficiency in a mice tumor model. Theis work can be published in Molecules after addressing the following issues.

Comments:

1.    The biodistribution study was conducted only in one mouse in Figure 9. The heterogeneity widely exists in animals, and it is generally conducted in a group of nice repeatedly (n = 3 at least).

2.    The singlet oxygen generation efficiency was suggested to be measured to compare free Ce6 with nanoparticles.

3.    The ratio for SU and Ce6 were 6/7 and 1 respectively. Why? Any reasons for the design.

Author Response

This manuscript by Jeong et al reports the preparation of CD-SS-Ce6 conjugate, and the photophysical properties and applications after self-assembling into nanoparticles. These nanoparticles showed redox-responsive function and enhanced therapeutic efficiency in a mice tumor model. These work can be published in Molecules after addressing the following issues.

Comments:

  1. The biodistribution study was conducted only in one mouse in Figure 9. The heterogeneity widely exists in animals, and it is generally conducted in a group of nice repeatedly (n = 3 at least).

Answer) Thanks for your valuable comment. Practically, we wanted to approve the reason for higher PDT efficacy of bCDsu-ss-Ce6 nanoparticles compared to free Ce6 in vivo animal study. From these reasons, we adapted animal imaging study with one animal to prove in vivo PDT efficacy. Then, we used one mouse for each treatment and, furthermore, committee of animal care and use recommended to minimize usage of animals for experiment. Thanks again for your comment.

  1. The singlet oxygen generation efficiency was suggested to be measured to compare free Ce6 with nanoparticles.

Answer) Thanks for your valuable comment. Practically, we already measured singlet oxygen (SO) efficiency but SO generation from bCDsu-ss-Ce6 was measured. Practically, at cell culture study, (as shown in Figure 7(b),) intracellular ROS generation of bCDsu-ss-Ce6 nanoparticles was not significantly increased rather than that we expected. Because GSH is an antioxidant molecule, GSH may compete with ROS in intracellular level. Then, we are going to do the study on the effect of GSH, ROS (H2O2) and SO on the PDT efficacy, degradation characteristics and physicochemical properties of bCDsu-ss-Ce6 conjugates in vitro and/or cell culture study. We will report these studies in another article in the near future. Thanks for your comment again.

SO generation from Ce6 alone or bCDsu-ssCe6 nanophotosensitizers (bCDsu-ssCe6 NP) with or without light irradiation at 664 nm.

  1. The ratio for SU and Ce6 were 6/7 and 1 respectively. Why? Any reasons for the design.

Answer) Thanks for your valuable comment. Practically, Ce6 is a hydrophobic molecule and has low solubility in water. One of the aim of this work is to make water-soluble chlorin e6 to handle easily. In our practical synthesis experience, higher Ce6 contents resulted in lower solubility in aqueous solution. Therefore, we decided that 1 Ce6 for 1 bCDsu is better to solubilize it. The reason for 6/7 SU for bCDsu is that higher density of SU moiety is helpful to react with amine-functional molecules. In future research, we will attach multiple molecules to the SU groups in bCDsu conjugates for cancer targeting and report in the other manuscript. Thanks for your comment.
